# Predicting the Mine Friction Coefficient Using the GSCV-RF Hybrid Approach

**Chenyang Guo [1], Xiaodong Wang [2,*], Dexing He [2], Jie Liu [1], Hongkun Li [2], Mengjiao Jiang [1] and Yu Zhang [1]**

[1] Faculty of Public Security and Emergency Management, Kunming University of Science and Technology, Kunming 650093, China
[2] Faculty of Land Resources Engineering, Kunming University of Science and Technology, Kunming 650093, China
* Correspondence: angiaoongwxd@163.com

**Abstract:** The safety and reliability of a ventilation system relies on an accurate friction resistance coefficient ($\alpha$), but obtaining $\alpha$ requires a great deal of tedious measurement work in order to determine the result, and many erroneous data are obtained. Therefore, it is vital that $\alpha$ be obtained quickly and accurately for the ventilation system design. In this study, a passive and active support indicator system was constructed for the prediction of $\alpha$. An RF model, GSCV-RF model and BP model were constructed using the RF algorithm, GSCV algorithm and BP neural network, respectively, for $\alpha$ prediction. In the GSCV-RF and BP models, 160 samples complied with the prediction indicator system and were used to construct a prediction dataset and, this dataset was divided into a training set and a test set. The prediction results were based on the quantitative evaluation models of *MAE*, *RMSE* and $R^2$. The results show that, among the three models, the GSCV-RF model's prediction result for $\alpha$ was the best, the RF model performed well and the BP model performed worst. In the prediction for all the datasets obtained by GSCV-RF model, all the values of *MAE* and *RMSE* were less than 0.5, the values of $R^2$ were more than 0.85 and the value of $R^2$ of the passive and active support test sets were 0.8845 and 0.9294, respectively. This proved that the GSCV-RF model can offer a more accurate $\alpha$ and aid in the reasonable design and the safe operation of a ventilation system.

**Keywords:** safety engineering; mine friction resistance coefficient; random forest; GSCV-RF; roadway support

## 1. Introduction

The coefficient of frictional resistance ($\alpha$) is an essential parameter for the calculating mine ventilation resistance, solving the ventilation network and optimizing the ventilation systems. The main method of obtaining this parameter is field measurement, but this method incurs a heavy workload that is also detailed and complicated and is easily affected by the operator, the equipment or the measurement method, leading to measurement result errors [1–4]. In addition, with the advancement of ore body mining, the mining sites gradually become deeper, so that it is impossible to carry out survey work on the tunnels that are in the planning stage and not constructed, something which can also result in missing data. All these problems may affect the study of ventilation systems and reduce the safety and reliability of the system. Therefore, obtaining $\alpha$ more quickly, accurately, and easily is a valuable research objective within ventilation system studies.

To solve this problem, some scientific researchers began with data mining. Shao [3] collected all the historical $\alpha$, constructed an $\alpha$ database, and matched the satisfied $\alpha$ through a fuzzy query for the ventilation system design of roadways without resistance measurement work. Liang et al. [5] introduced more detailed measurement indicators during the construction of their $\alpha$ database, which can match the corresponding $\alpha$ more accurately and precisely with the roadway and further improve the security of the ventilation system.

However, the data mining method requires a large amount of detailed measurement data with rich measurement indicators in order to ensure the accuracy of the matched data. Therefore, collecting enough data for this method is a great challenge.

Considering the data mining defects, scientific researchers have adopted machine learning, as it has a lower cost and requires less time to solve the detailed measurement problems and obtain reliable data quickly and easily [6]. Zhang et al. [7] started with the use of a back-propagation (BP) neural network to predict the $\alpha$ of log-supported roadways, which provided a new method for obtaining an accurate and reliable $\alpha$. Wang [8] then followed the BP neural network. They started with a type of roadway support mode and constructed an $\alpha$ prediction model for a variety of roadway support modes. This ensured that the $\alpha$ prediction of the BP neural network was no longer limited to a certain type of roadway. Wei [9] introduced the parameter of the cross-section shape of a roadway, optimizing the $\alpha$ prediction model through the BP neural network and making more accurate $\alpha$ predictions through the model. Most machine learning models used for predicting $\alpha$ have been developed based on BP neural networks, but BP neural networks have disadvantages. These include the tendency of falling into a local minimum value, which leads to training failure and overfitting. Therefore, it is necessary to spend time adjusting the prediction model so as to ensure the prediction result accuracy [10,11].

In addition to BP neural networks, there are many other machine learning methods with different characteristics. Breiman [12] proposed the random forest (RF) algorithm, which is superior in handling regression problems through the examination of detailed examples. The algorithm has the advantages of requiring fewer tuning parameters, having a higher training efficiency and requiring less overfitting than the BP neural network [13,14]. For the prediction of the $\alpha$ regression problem, the RF is also a solution method. Li et al. [15] followed the variety roadway prediction indicator system of [8] and constructed a variety of RF prediction models of $\alpha$ and achieved better prediction results. However, there is still room for improvement.

Therefore, considering the influences of the values of the hyperparameters on the RF prediction results, in this paper, we optimize an RF algorithm with a GSCV algorithm to construct a GSCV-RF $\alpha$ prediction model. This can obtain a more accurate $\alpha$ prediction and solve problems such as the detailed and complicated measurement work, large measurement errors and frequent missing data, problems that frequently affect the accuracy of ventilation system studies. The detailed workflow can be seen in Figure 1.

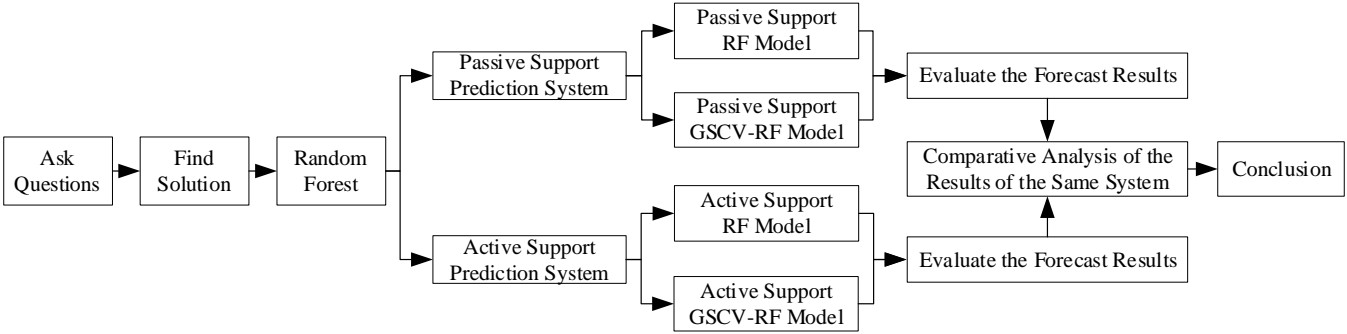

**Figure 1.** Workflow of this paper.

## 2. Predictive Model Construction

### 2.1. RF Prediction Models

RF offers a new solution for the indirect resolution of classification and regression parameters [16]. This method has the advantages of requiring fewer tuning parameters, a high training efficiency and less susceptibility to overfitting. The regression problem is handled by building multiple unrelated CART decision trees with decreasing computational accuracy, and the output values of all the trees are averaged as the output of RF [17–19]. The algorithm used for handling regression problems of the prediction of $\alpha$ is as follows [20,21]:

1.  Input conditions for forest growth: RF prediction results are heavily influenced by three hyperparameters: the number of decisions, the maximum number of features and the maximum depth of the decision tree, which are, respectively, defined as $x_1$, $x_2$ and $x_3$ and used as the input growth conditions $(x_1, x_2, x_3)$.

2.  The dataset containing $N$ samples with $O$ input features is sampled $N$ times using put-back sampling, and $o$ features are selected randomly to serve as the input features. This process is repeated $N$ times to generate $N$ training datasets including $N$ samples with $o$ input features $(o \leq O)$:

$$\begin{aligned} D_1 &= x_{11}, y_{11}, (x_{12}, y_{12}), \cdots, (x_{1N}, y_{1N}) \\ D_2 &= x_{21}, y_{21}, (x_{22}, y_{22}), \cdots, (x_{2N}, y_{2N}) \\ &\qquad\qquad \cdots \\ D_N &= x_{N1}, y_{N1}, (x_{N2}, y_{N2}), \cdots, (x_{NN}, y_{NN}) \end{aligned} \tag{1}$$

where $D_N$ is the $N$th dataset of training numbers; $x_{NN}$ is the input data under the $N$th sample of the $N$th training dataset; and $y_{NN}$ is the output data under the $N$th sample of the $N$th training dataset.

Among them:

$$x_{NN} = (x_{NN1}, x_{NN2}, \cdots, x_{NNj}) \tag{2}$$

where $X_{NNj}$ is the $j$th input data under the $N$th sample in the $N$th training dataset.

3.  Choose one of the datasets, select the appropriate cut variable $j$ and cut point $s$, and ensure the segmentation effect using Equation (3).

$$\min_{j,s} \left[ \min_{c_1} \sum_{x_i \in R_1(j,s)} (y_i - c_1)^2 + \min_{c_2} \sum_{x_i \in R_2(j,s)} (y_i - c_2)^2 \right] \tag{3}$$

where $y_i$ is the output data for the $i$th sample in the dataset; $c_1$ is the mean of all $y_i$ under the partitioned region $R_1$; and $c_2$ is the mean of all $y_i$ under the partitioned region $R_2$.

4.  The optimal $(j, s)$ is partitioned into regions to obtain $R_1$ and $R_2$, and the output value of the corresponding region $\hat{c}_m$ is determined:

$$\begin{aligned} R_1(j,s) &= \left\{ x \big| x^{(j)} \leq s \right\}, \quad R_2(j,s) = \left\{ x \big| x^{(j)} > s \right\} \\ \hat{c}_m &= \frac{1}{N_m} \sum_{x_i \in R_m(j,s)} y_i, \quad x \in R_m, \quad m = 1, 2 \end{aligned} \tag{4}$$

where $R_1$ and $R_2$ are the data region according to $(j, s)$; $x^{(j)}$ is the selected optimal division variable; and $N_m$ is the number of samples in the delimited region $R_m$.

5.  Until the requirements for the decision tree's growth are satisfied, repeat steps 2 and 3 for the divided subregions.

6.  To construct a decision tree, divide the input space into $M$ regions, $R_1$, $R_2$, $\cdots$, $R_M$.

$$f(x) = \sum_{m=1}^{M} \hat{c}_m I(x \in R_m) \tag{5}$$

where $f(x)$ is the resulting decision tree.

7.  Repeat steps 2, 3, 4 and 5 until the forest's growth requirements are satisfied and an equal number of decision trees are formed so as to form a random forest.

Figure 2 depicts a flowchart of the RF algorithm based on the preceding algorithm. Following these steps, a prediction model is formed by applying this to the prediction.

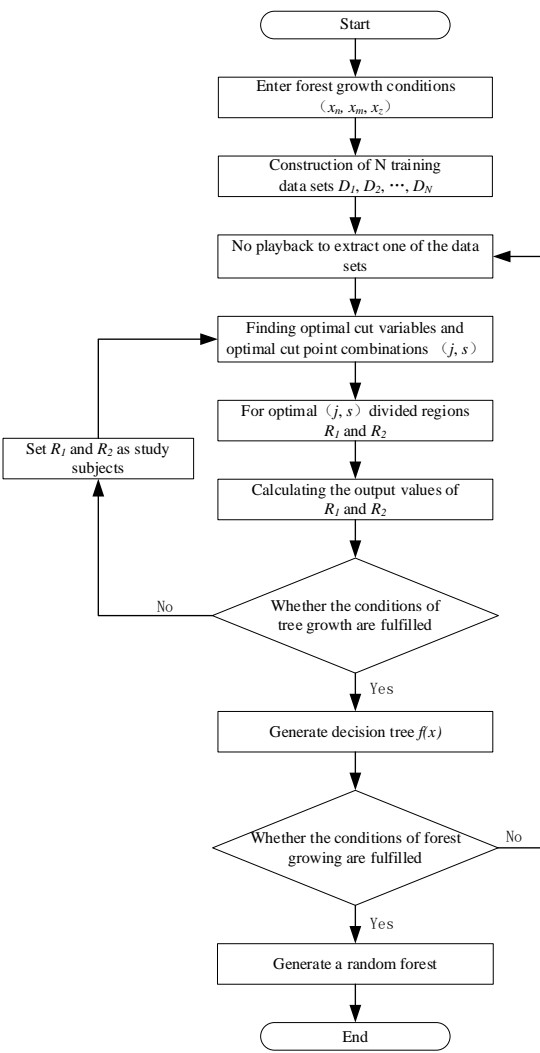

**Figure 2.** RF algorithm flow chart.

1.  Select the appropriate input and output properties to build a prediction indicator system.
2.  Use numerical and clustering methods to process the data such that it satisfies the RF model's requirements.
3.  Adopting a given percentage, divide the dataset into a training set for training the model and a test set for testing the prediction.
4.  Input the model parameters (growth conditions), such as the number of decisions, maximum number of features and maximum depth of the decision tree.
5.  Train the training set's $\alpha$ prediction model.
6.  Predict the $\alpha$ of the test set.
7.  According to the predictions to evaluation the constructed RF forecasting model. The evaluation indicators include mean absolute error (*MAE*), root mean square error (*RMSE*) and model goodness of fit ($R^2$):

$$MAE = \frac{1}{n}\sum_{i=1}^{n}|\hat{y}_i - y_i| \tag{6}$$

$$RMSE = \sqrt{\frac{1}{n}\sum_{i=1}^{n}|\hat{y}_i - y_i|^2} \tag{7}$$

$$R^2 = 1 - \frac{\sum_{i=1}^{n} (\hat{y}_i - y_i)^2}{\sum_{i=1}^{n} (\overline{y}_i - y_i)^2} \tag{8}$$

where $n$ is the test set sample size; $\hat{y}_i$ is the predicted value of the ith test set sample; $y_i$ is the true value of the ith test set sample; and $\overline{y}_i$ is the mean of the sample true values.

### 2.2. GSCV Optimization Algorithm

GSCV (grid search cross-validation) is an algorithm comprised of a grid search and cross-validation that enables automatic parameter tuning in order to identify the ideal combination of parameters, which frequently optimizes the process, in conjunction with other algorithms [22]. As the prediction results of RF are significantly affected by the values of the hyperparameters, the GSCV algorithm is introduced and combined with RF to circumvent this drawback. GSCV is then used to optimize RF by determining the optimal input parameters so as to build a GSCV-RF prediction model for predicting $\alpha$. The GSCV optimization algorithm is as follows:

1. Set the range of each hyperparameter and set range of RF's hyperparameter (growth condition) as an example:

$$\begin{aligned} x_1 &\in [1, n] \\ x_2 &\in [1, m] \\ x_3 &\in [1, z] \end{aligned} \tag{9}$$

   where $n$ is the upper limit of the value of the hyperparameter $x_1$; $m$ is the upper limit of the value of the hyperparameter $x_2$; and $z$ is the upper limit of the value of the hyperparameter $x_3$.

2. To obtain a hyperparameter combination, set each hyperparameter individually. Assuming that each hyperparameter step is 1, $n \times m \times z$ hyperparameter combinations $(x_n, x_m, x_z)$ are created. Hence, each hyperparameter combination represents one of the growth conditions of RF.

3. To avoid the chance of outcomes owing to dataset partitioning, the dataset is divided into $K$ mutually exclusive subsets of the same size, $d_1, d_2, \cdots, d_k$, and each subset is utilized as a separate validation set once, and the remaining $K$-1 subsets are used to produce $K$ new datasets.

4. Each hyperparameter combination $(x_n, x_m, x_z)$ is trained once on each of the $K$ new datasets, the goodness-of-fit $R_1^2, R_2^2, \cdots, R_k^2$ under each dataset is produced, and the output of the hyperparameter combination is the mean value $\overline{R^2_{(x_n, x_m, x_z)}}$ of the corresponding goodness-of-fit for each dataset.

$$\overline{R^2_{(x_n, x_m, x_z)}} = \frac{1}{k} \sum_{i=1}^{k} R_i^2 \tag{10}$$

5. Repeat step 4 for each combination of hyperparameters in order to identify the optimal output as an input parameter for the algorithm combined with GSCV. The following is an expression of the optimal output:

$$\max_{x_n, x_m, x_z} \left[ \overline{R^2_{(x_n, x_m, x_z)}} \right] \tag{11}$$

### 2.3. GSCV-RF Prediction Model

The GSCV algorithm is used to optimize the RF algorithm to produce the GSCV-RF algorithm, and the algorithm flow is depicted in Figure 3. The method was used to predict $\alpha$ and develop the GSCV-RF prediction model. The model's implementation phases are depicted in Figure 4. In the GSCV-RF model's optimization of the RF model, the inability to determine the input hyperparameters is addressed in five steps:

1.  The range and step size of the three hyperparameters, including the number of decision trees, the maximum number of features and the maximum decision tree depth, are established.
2.  Combining the values of each hyperparameter to individually yields all the possible hyperparameter combinations.
3.  The $\alpha$ dataset is divided into *K* equal parts, with *K-1* parts serving as the training set and the remaining 1 part serving as the test set. After *K* repetitions, each sample serves as one test set, resulting in *K* new datasets.
4.  Using the new dataset, each combination of hyperparameters is subjected to K-fold cross-validation.
5.  The results produced for each hyperparameter combination are scored, and the combination with the highest score is used as the model's input parameter.

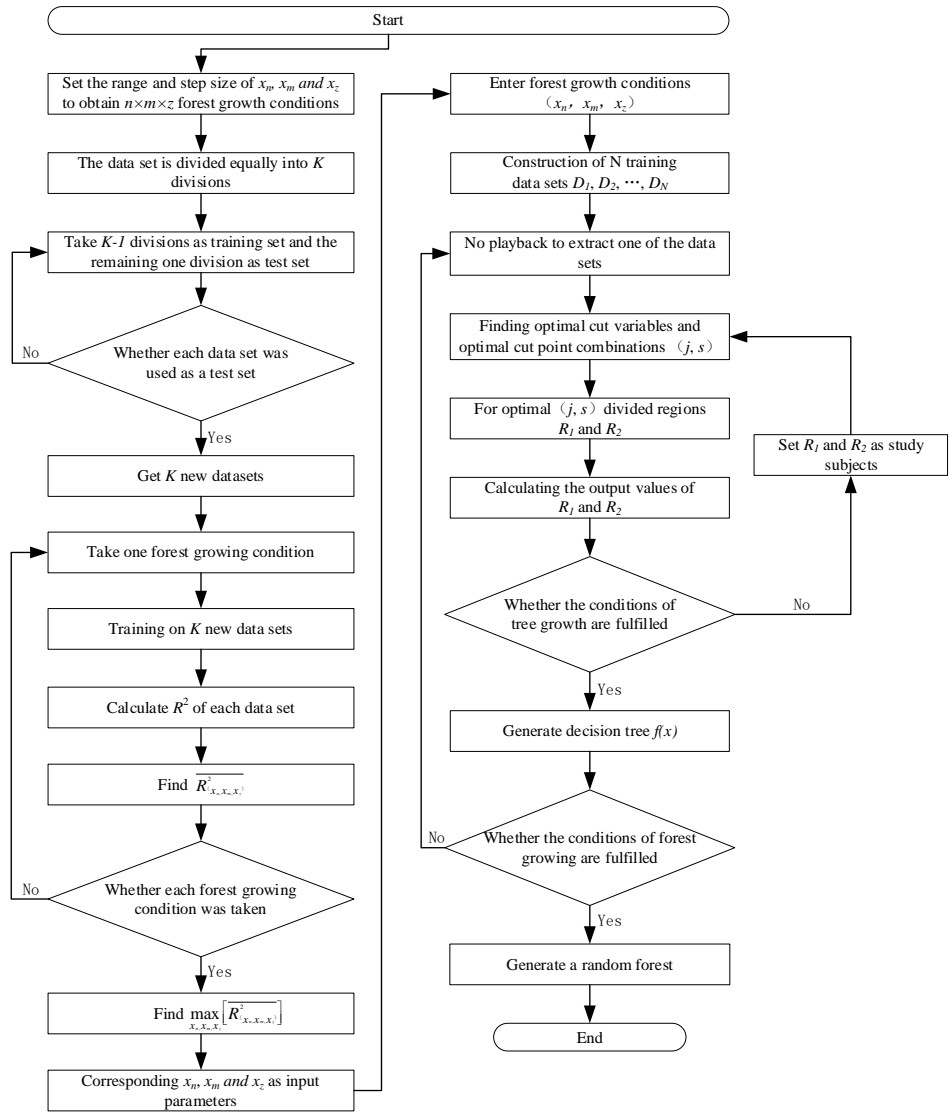

**Figure 3.** GSCV-RF algorithm flow chart.

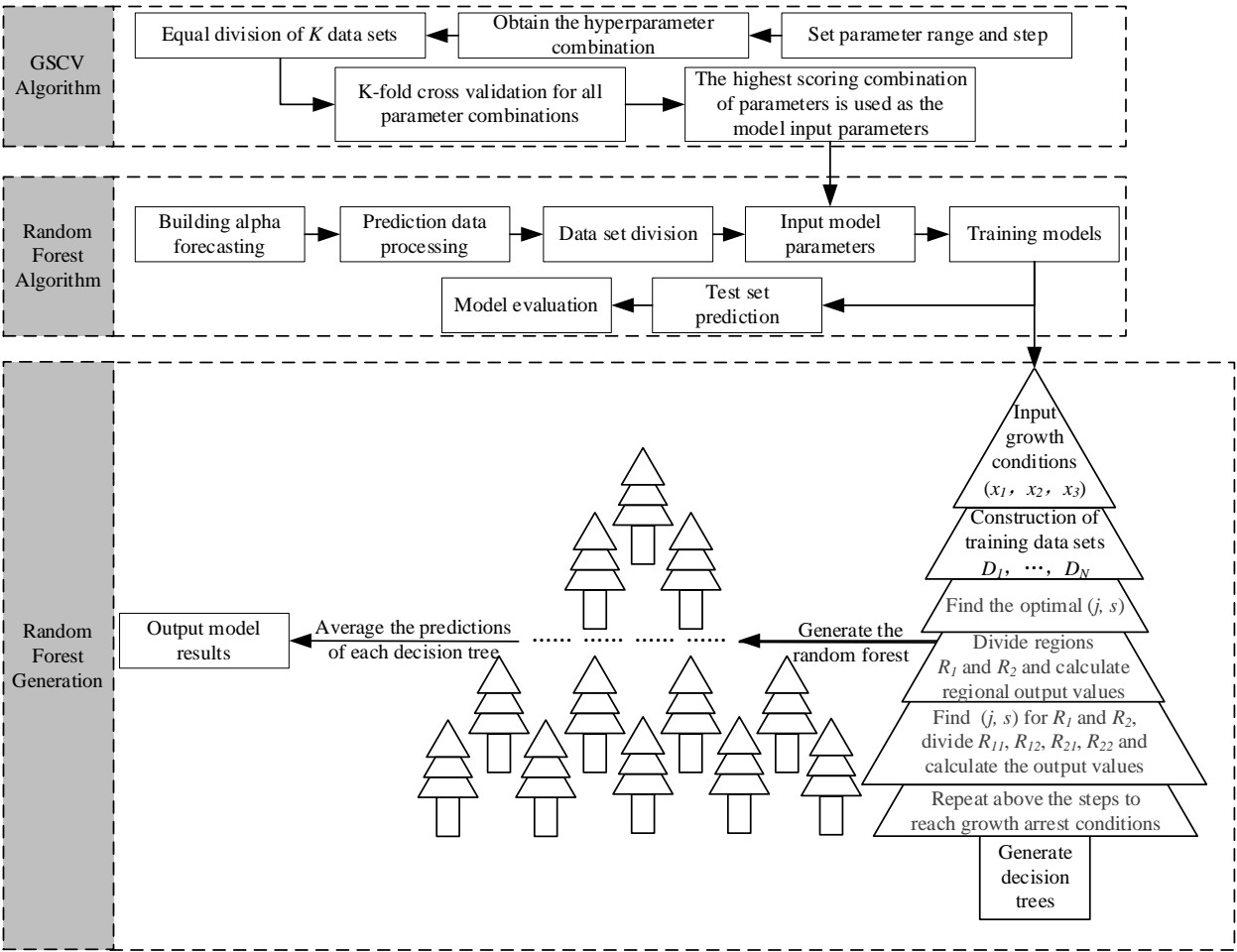

**Figure 4.** GSCV-RF model prediction process.

## 3. Example Analysis

### 3.1. Constructing a Forecasting Indicator System

The $\alpha$ has a close relationship with the roughness of the shaft wall, which is affected by the various support mode types. The passive support mode, characterized by bracket support, and active support mode, characterized by bolt support, are the two most prevalent support modes utilized underground [23]. Currently, two support modes are in regular usage. The type of roadway support chosen is dictated by the lithology of the subsurface strata, the depth of the mining and other factors. Consequently, the samples used to predict $\alpha$ are separated into passive support and active support samples according to the support mode employed for the roadway. Combining the findings on $\alpha$ prediction found in the literature [8,24], the passive and active support indicator systems for predicting $\alpha$ are correspondingly created (see Figure 5).

### 3.2. Data Selection and Processing

The authors of [8,24,25] each conducted a study investigating how to determine $\alpha$, but there were too many data listed, which was not compatible with the $\alpha$ prediction indicator system constructed in Section 3.1. Consequently, we must select the appropriate data as the research sample for the paper. Two types of dataset in the literature [8] were related to the paper's passive support $\alpha$ prediction indicator system. We selected 50 sets of data from each of the two types of data training sets and their test data sets for a total of 124 sets of data used to construct the passive support prediction dataset. Some research data in the literature [24] were related to the active support $\alpha$ prediction indicator system of this paper, and we incorporated all of them into our research. However, there were only 22 sets of

data in total. Thus, we needed to add more related data from the literature [25] to construct an active support prediction dataset with a total of 36 sets of data. In the end, 160 sets of research samples were included in the publication.

According to data type classification, the data type of each dataset indicator can be categorized as the numeric or character type, as shown in Table 1. The RF prediction model and the GSCV-RF prediction model require the data of the samples to be of the numeric type. Thus, the character-based data of the "Support Type" and "Cross-section Profile" are numbered with the numbers 1, 2, . . . , *n* to represent the various support types and section shapes, respectively. This procedure is used to process the data.

The prediction dataset is then divided into a training set for the training model and a test set for the prediction of the effect of the test model. In our study, there were 124 samples of passive support prediction datasets, of which 80% were training sets and 20% were test sets, and there were 36 samples of active support prediction datasets. As the quantity was too small, to improve the accuracy of the prediction model, we increased the training sets ratio so that 85% of the samples were training sets and 20% were test sets. In addition, we corrected the division of the datasets, so that the data used in the training set and test set were identical in the subsequent model.

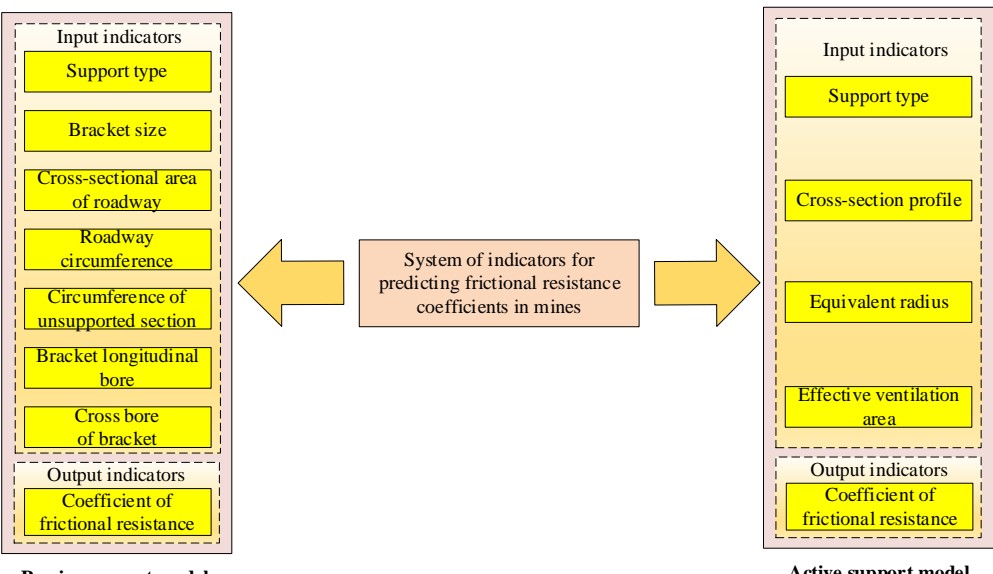

**Figure 5.** System of indicators for predicting frictional resistance coefficients in mines.

**Table 1.** Data categories of the study sample.

| No. | Indicator | Data Type |
|---|---|---|
| 1 | Support Type | Character type |
| 2 | Cross-Section Profile | Character type |
| 3 | Bracket Size | Numerical type |
| 4 | Roadway Cross-Sectional Area | Numerical type |
| 5 | Lane Circumference | Numerical type |
| 6 | Perimeter of Unsupported Section | Numerical type |
| 7 | Bracket Longitudinal Bore | Numerical type |
| 8 | Cross-Bore of Bracket | Numerical type |
| 9 | Equivalent Radius | Numerical type |
| 10 | Effective Ventilation Area Factor | Numerical type |
| 11 | Coefficient of frictional resistance | Numerical type |

### 3.3. Data Statistics

Figures 6 and 7 depict violin plots, which represent the statistics of the paper's research samples. In Tables 2 and 3, showing the statistical indicators of all the training set and test set calculations, the statistical indicators are provided.

**Table 2.** Table of the parameter statistics of the passive support dataset.

| Indicators | Min. | Max. | Avg. | St. D. | Med. | S. Var. | St. E. | Kurt. | Skew. | Range | Mode |
|---|---|---|---|---|---|---|---|---|---|---|---|
| Training Datasets | | | | | | | | | | | |
| Support Type | 1 | 2 | 1.5 | 0.503 | 1.5 | 0.253 | 0.05 | −2.041 | 0 | 1 | |
| Bracket Size | 10 | 26 | 15.68 | 4.126 | 15 | 17.028 | 0.413 | 0.002 | 0.592 | 16 | 10 |
| Roadway Cross-Sectional Area | 4 | 10 | 6.94 | 2.247 | 6 | 5.047 | 0.225 | −1.366 | 0.033 | 6 | 4 |
| Lane Circumference | 8.32 | 13.16 | 10.812 | 1.816 | 10.19 | 3.297 | 0.182 | −1.358 | −0.13 | 4.84 | 8.32 |
| Perimeter of Unsupported Section | 2.13 | 3.37 | 2.712 | 0.509 | 2.61 | 0.259 | 0.051 | −1.668 | 0.021 | 1.24 | 2.13 |
| Bracket Longitudinal Bore | 3 | 8 | 4.97 | 1.85 | 5 | 3.423 | 0.185 | −1.218 | 0.416 | 5 | 3 |
| Cross-Bore of Bracket | 0.033 | 0.135 | 0.065 | 0.021 | 0.062 | 0 | 0.002 | 0.69 | 0.816 | 0.102 | 0.059 |
| $\alpha$ | 0.071 | 0.261 | 0.121 | 0.042 | 0.106 | 0.002 | 0.004 | 1.049 | 1.291 | 0.19 | 0.137 |
| Testing Datasets | | | | | | | | | | | |
| Support Type | 1 | 2 | 1.667 | 0.482 | 2 | 0.232 | 0.098 | −1.568 | −0.755 | 1 | 2 |
| Bracket Size | 10 | 24 | 15.708 | 4.059 | 16 | 16.476 | 0.829 | −0.294 | 0.32 | 14 | 16, 18 |
| Roadway Cross-Sectional Area | 4 | 10 | 6.542 | 1.865 | 6 | 3.476 | 0.381 | −0.927 | 0.39 | 6 | 5 |
| Lane Circumference | 8.32 | 13.16 | 10.538 | 1.51 | 10.19 | 2.281 | 0.308 | −1.058 | 0.196 | 4.84 | 9.30 |
| Perimeter of Unsupported Section | 2.13 | 3.37 | 2.702 | 0.386 | 2.61 | 0.149 | 0.079 | −1.052 | 0.188 | 1.24 | 2.39 |
| Bracket Longitudinal Bore | 3 | 8 | 4.75 | 1.622 | 4 | 2.63 | 0.331 | −0.049 | 0.976 | 5 | 4 |
| Cross-Bore of Bracket | 0.035 | 0.094 | 0.066 | 0.018 | 0.066 | 0 | 0.004 | −1.347 | −0.164 | 0.059 | 0.084 |
| $\alpha$ | 0.092 | 0.273 | 0.134 | 0.047 | 0.118 | 0.002 | 0.01 | 1.735 | 1.407 | 0.181 | 0.916 0.118 0.143 |

**Table 3.** Table of the parameter statistics of the active support dataset.

| Indicators | Min. | Max. | Avg. | St. D. | Med. | S. Var. | St. E. | Kurt. | Skew. | Range | Mode |
|---|---|---|---|---|---|---|---|---|---|---|---|
| Training Datasets | | | | | | | | | | | |
| Support Type | 1 | 2 | 1.367 | 0.49 | 1 | 0.24 | 0.089 | −1.784 | 0.583 | 1 | 1 |
| Cross-Section Profile | 1 | 2 | 1.433 | 0.504 | 1 | 0.254 | 0.092 | −2.062 | 0.283 | 1 | 1 |
| Equivalent Radius | 0.828 | 2.3 | 1.389 | 0.505 | 1.158 | 0.255 | 0.092 | −1.372 | 0.539 | 1.472 | 0.835 2.000 |
| Effective Ventilation Area Factor | 0.84 | 1 | 0.954 | 0.044 | 0.96 | 0.002 | 0.008 | 0.927 | −1.122 | 0.16 | 1.00 |
| $\alpha$ | 0.01 | 0.045 | 0.021 | 0.01 | 0.017 | 0 | 0.002 | −0.139 | 0.91 | 0.035 | 0.029 |
| Testing Datasets | | | | | | | | | | | |
| Support Type | 1 | 2 | 1.333 | 0.516 | 1 | 0.267 | 0.211 | −1.875 | 0.968 | 1 | 1 |
| Cross-Section Profile | 1 | 2 | 1.5 | 0.548 | 1.5 | 0.3 | 0.224 | −3.333 | 0 | 1 | |
| Equivalent Radius | 0.884 | 2.05 | 1.283 | 0.482 | 1.048 | 0.232 | 0.197 | −0.697 | 1.064 | 1.166 | |
| Effective Ventilation Area Factor | 0.87 | 1 | 0.962 | 0.047 | 0.975 | 0.002 | 0.019 | 4.225 | −1.967 | 0.13 | |
| $\alpha$ | 0.01 | 0.042 | 0.023 | 0.012 | 0.02 | 0 | 0.005 | −0.089 | 0.802 | 0.032 | |

### 3.4. RF Model Prediction

Using the passive and active support $\alpha$ prediction indicator system and Python, the passive and active support RF prediction models were developed. The input parameters of the models were default parameters (see Table 4 for details). After the training, the passive and active support training set models were used to predict $\alpha$ for the associated test sets, and the results are depicted in Figures 8 and 9.

As demonstrated in Figures 8a and 9a, the majority of the samples were positioned near y = x. Hence, the actual measurement value of the passive and active support training sets and test sets was more closely aligned with the prediction value. In the passive support training sets, the values of *MAE*, *RMSE* and $R^2$ were 0.0019, 0.0029 and 0.9952, respectively, and in the passive support test sets, the values were 0.0112, 0.0159 and 0.8814, respectively. In the active support training sets, the values were 0.0010, 0.0015 and 0.9775, respectively, and in the active support test sets, the values were 0.0027, 0.0031 and 0.9165, respectively. All the values of the *MAE* and *RMSE* of the datasets were less than 0.05, and the value of $R^2$ was more than 0.85.

As observed in Figures 8b,c and 9b,c, the sample numbers of the passive support test sets were 1, 2, 6, 11, 14, 15, 18, 21 and 24. The deviation of the prediction from the field-measured value was slightly large, accounting for 37.5% of all the sample numbers. The sample numbers of the active support training sets were 1, 8, 11, 23 and 30. The deviation of the prediction value from the field-measured value was slightly large, accounting for 16.67% of all the sample numbers. The sample numbers of the active support test sets were 1, 2, 5 and 6. The deviation of the prediction from the field-measured value was slightly large, accounting for 66% of all the sample numbers. In addition, the predicted values of the remaining samples were close to the actual value.

In Figures 8d and 9d, the frequency distribution of the samples' relative error is depicted. Thus, we can conclude that the majority of the sample relative errors were closer to 0.

In conclusion, the RF model achieved superior prediction results for both the passive and active support datasets. Furthermore, the trained RF model was applied to the test set prediction, and the prediction result for the active support test sets was superior to that of the passive support test sets.

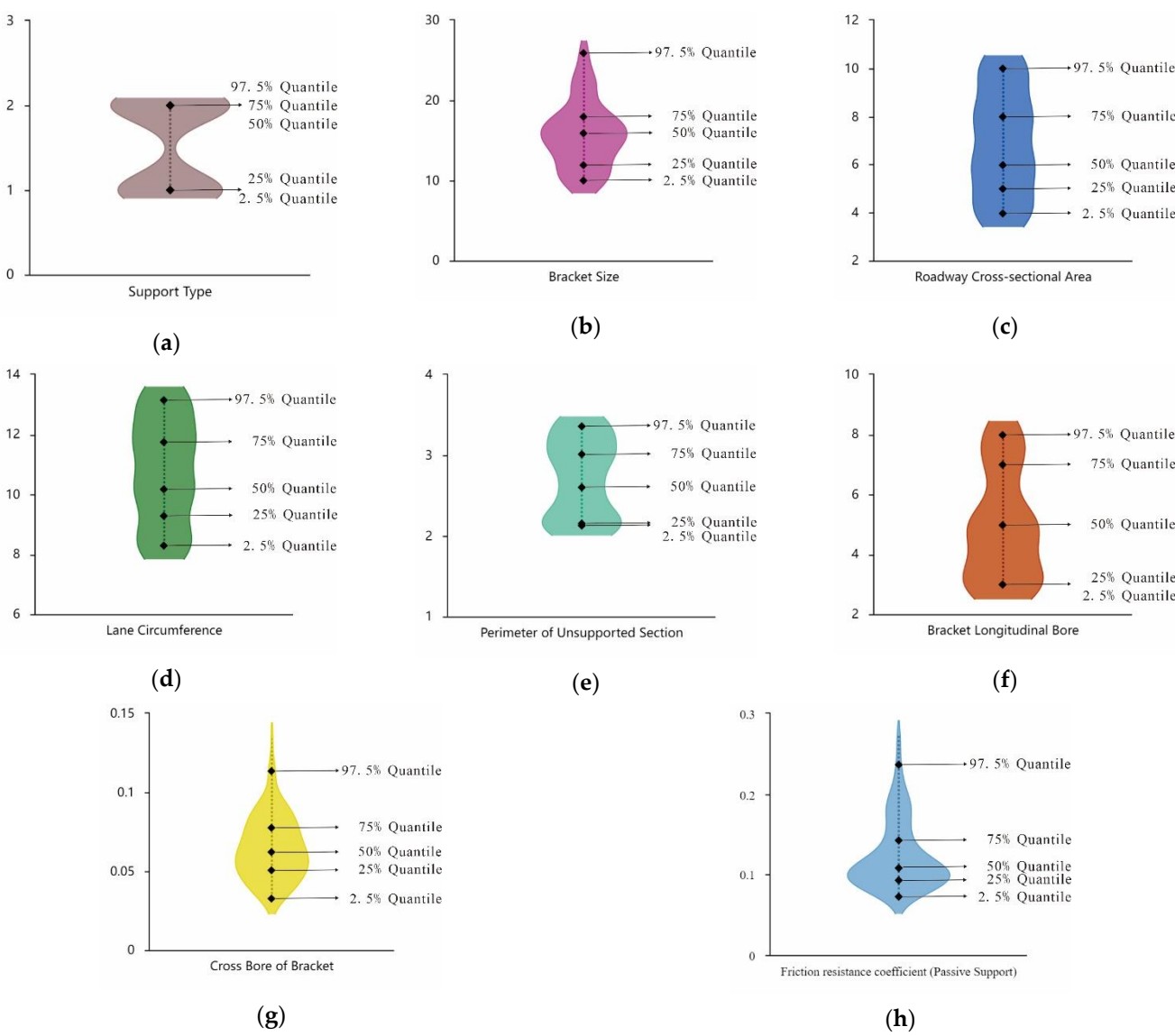

**Figure 6.** Violin plots showing the distribution of the passive support dataset. (**a**) Violin plot of the support-type data; (**b**) violin plot of the bracket size data; (**c**) violin plot of the roadway cross-sectional area data; (**d**) violin plot of the lane circumference data; (**e**) violin plot of the perimeter of the unsupported section data; (**f**) violin plot of the bracket longitudinal bore data; (**g**) violin plot of the cross-bore of bracket data; (**h**) violin plot of the friction resistance coefficient (passive support) data.

**Table 4.** Input parameters of the RF model.

| No. | Parameter Name | Value (Passive Support) | Value (Active Support) |
|-----|----------------|--------------------------|-------------------------|
| 1 | *n_estimators* | 100 | 100 |
| 2 | *max_features* | Auto | Auto |
| 3 | *max_depth* | None | None |

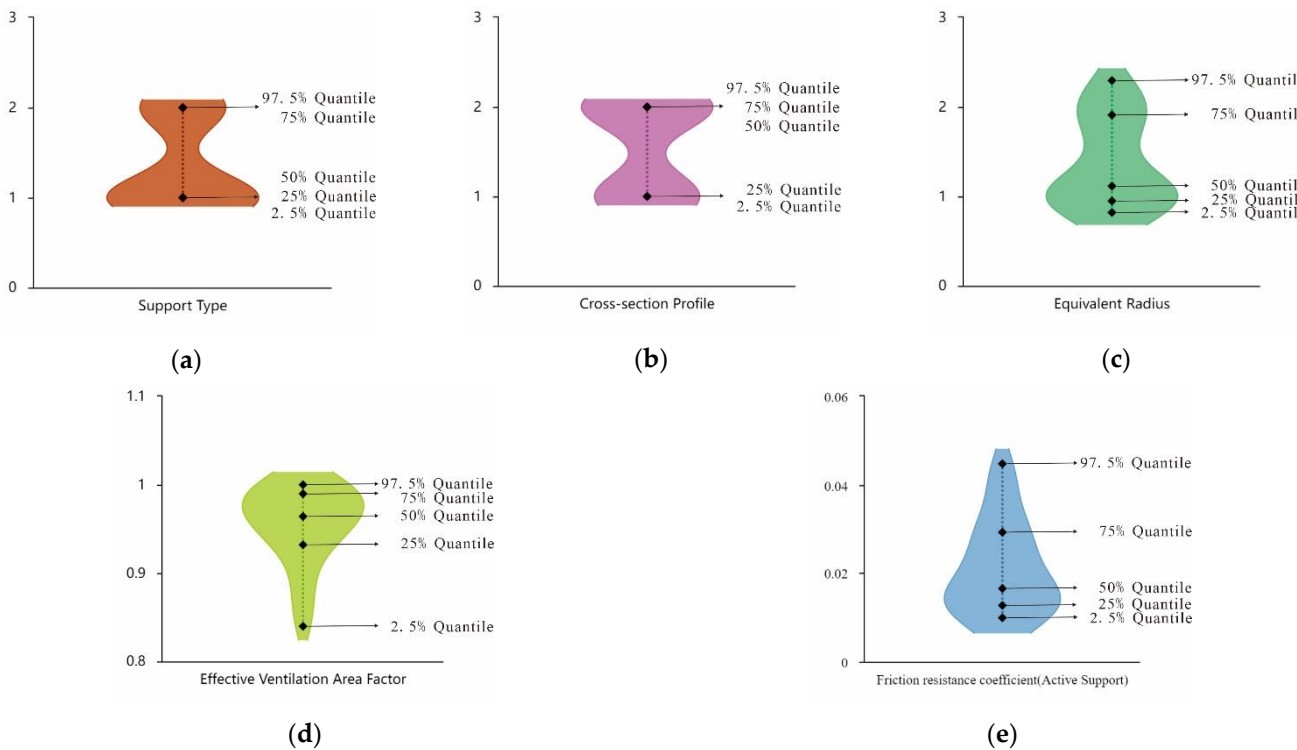

**Figure 7.** Violin plots showing the distribution of the active support dataset. (**a**) Violin plot of the support-type data; (**b**) violin plot of the cross-section profile data; (**c**) violin plot of the equivalent radius data; (**d**) violin plot of the effective ventilation area factor data; (**e**) violin plot of the perimeter of friction resistance coefficient (active support) data.

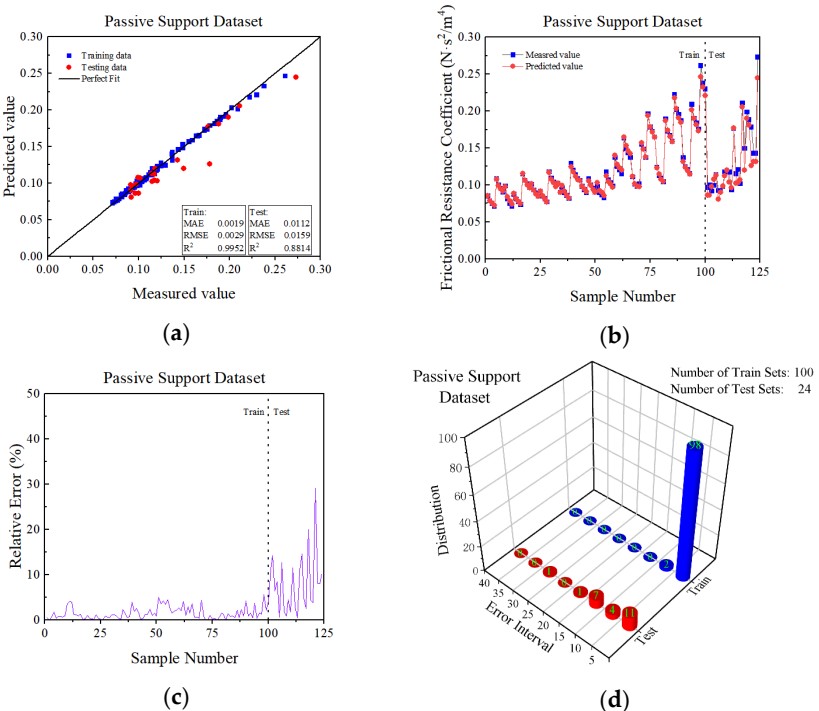

**Figure 8.** The passive support RF model prediction results. (**a**) Correlation evaluation of the measured and predicted values of $\alpha$; (**b**) curves of the measured and predicted values of the samples; (**c**) sample prediction error curve; (**d**) sample error frequency distribution.

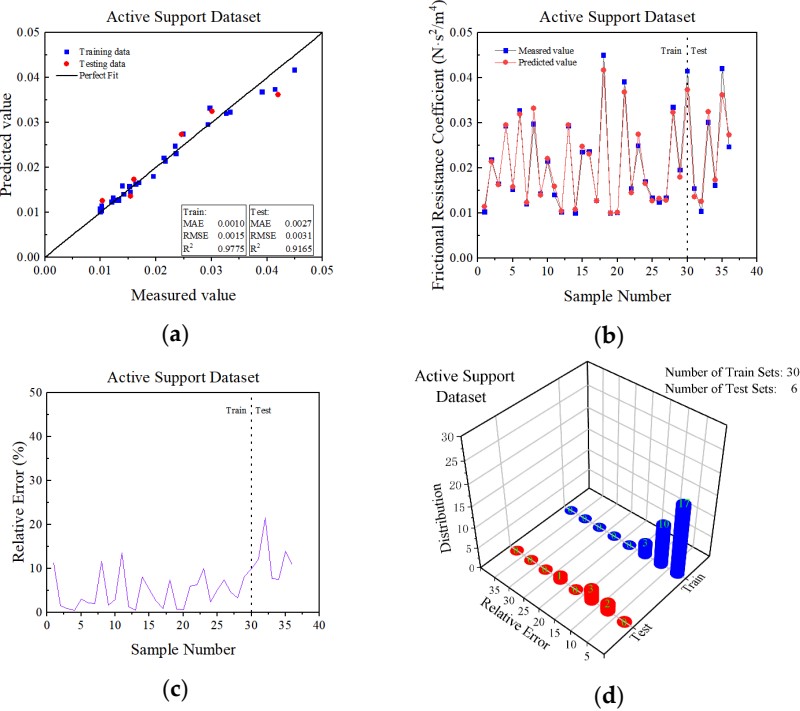

**Figure 9.** The active support RF model prediction results. (**a**) Correlation evaluation of the measured and predicted values of α; (**b**) curves of the measured and predicted values of the samples; (**c**) sample prediction error curve; (**d**) sample error frequency distribution.

### 3.5. GSCV-RF Model Predictions

#### 3.5.1. Searching for the Optimal Input Parameters

The passive and active support α prediction indicator systems and Python were utilized to build the passive and active support GSCV-RF prediction models. We set the value range and step size for each input parameter, as shown in Table 5, and set CV = 5 in order to identify the optimal parameter combination. The passive and active support prediction datasets were used for the parameter optimization samples, and the results of the optimization are shown in Table 6.

**Table 5.** GSCV-RF Model Parameter Optimization Settings.

| No. | Parameter Name | Parameter Range | Step Length |
|-----|----------------|-----------------|-------------|
| 1 | *n_estimators* | [10, 150] | 1 |
| 2 | *max_features* | [0.1, 1.0] | 0.1 |
| 3 | *max_depth* | [3, 50] | 1 |

**Table 6.** GSCV-RF Model Parameter Optimization Results.

| No. | Parameter Name | Optimum Value (Passive Support) | Optimum Value (Active Support) |
|-----|----------------|--------------------------------|-------------------------------|
| 1 | *n_estimators* | 67 | 110 |
| 2 | *max_features* | 0.9 | 1.0 |
| 3 | *max_depth* | 9 | 3 |

#### 3.5.2. Model Training and Prediction

The parameters shown in Table 6 were used as the input parameters, and training models for the passive and active support training sets were employed. Figures 10 and 11 depict the α of the corresponding test set prediction result after the training.

As demonstrated in Figures 10a and 11a, the majority of samples were positioned near y = x. Hence, we know that the actual measurement value of the passive and active support training sets and test sets was more closely comparable to the predicted value. In the passive support training sets, the values of *MAE*, *RMSE* and $R^2$ were 0.0018, 0.0025 and 0.9965, respectively, and in the passive support test sets, the values were 0.0112, 0.01597 and 0.8845, respectively. In the active support training sets, the values were 0.0014, 0.0019 and 0.9641, respectively, and in the active support test sets, the values were 0.0024, 0.0028 and 0.9294, respectively. All the values of the *MAE* and *RMSE* of datasets were less than 0.05, and the value of $R^2$ was more than 0.85.

As observed in Figure 10b,c and Figure 11b,c, the sample numbers of the passive support test sets were 1, 2, 6, 11, 14, 15, 18, 21 and 24. The deviation of the prediction from the field-measured value was slightly large, accounting for 37.5% of all the sample numbers. The sample numbers of the active support training sets were 1, 8, 11, 14, 22, 23, 26, 29 and 30. The deviation of the prediction from the field-measured value was slightly large, accounting for 30% of all the sample numbers. The sample numbers of the active support test sets were 1, 2, 5 and 6. The deviation of the prediction from the field-measured value was slightly large, which was 66% of all the sample numbers. In addition, the prediction values of the remaining samples were close to the actual value.

In Figures 10d and 11d, the frequency distribution of the samples' relative error is depicted. Thus, we can conclude that the majority of the samples' relative error was closer to 0.

As stated previously, the GSCV-RF model can achieve superior prediction results for both passive and active support datasets. Additionally, the trained GSCV-RF model was used for the test set prediction, and the prediction result of the active support test sets was superior to that of the passive support test sets.

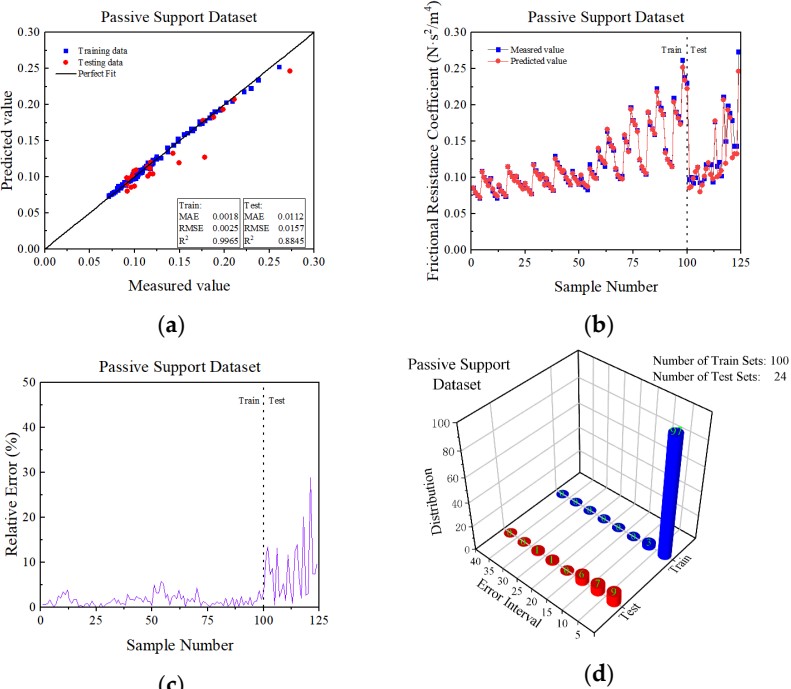

**Figure 10.** The passive support GSCV-RF model prediction results. (**a**) Correlation evaluation of the measured and predicted values of $\alpha$; (**b**) curves of the measured and predicted values of the samples; (**c**) sample prediction error curve; (**d**) sample error frequency distribution.

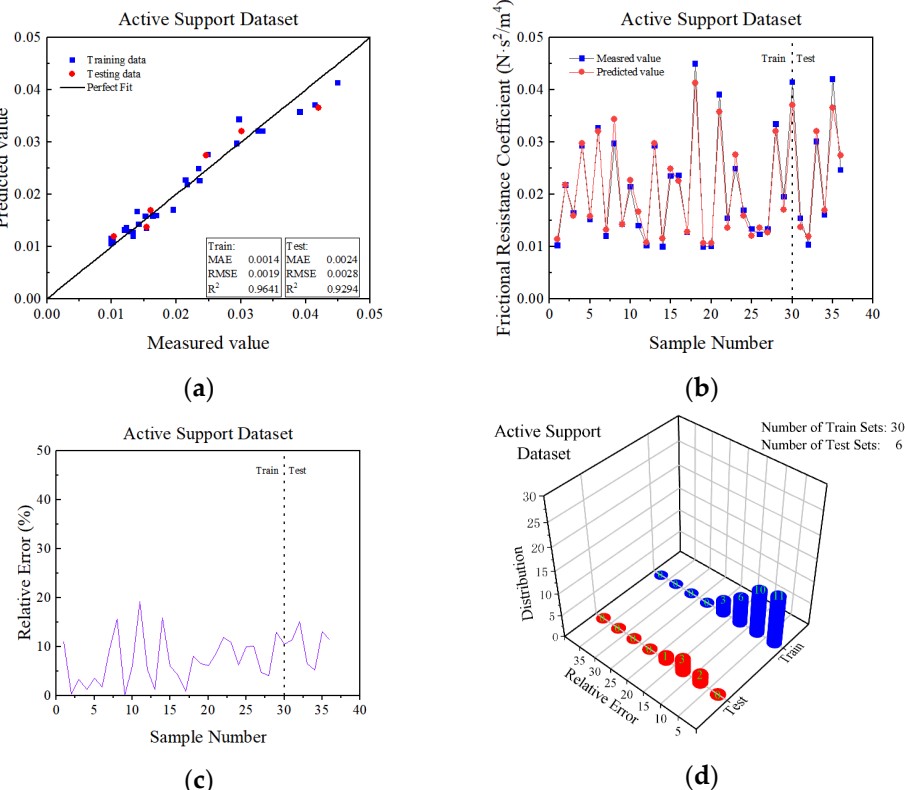

**Figure 11.** The active support GSCV-RF model prediction results. (**a**) Correlation evaluation of the measured and predicted values of $\alpha$; (**b**) curves of the measured and predicted values of the samples; (**c**) sample prediction error curve; (**d**) sample error frequency distribution.

### 3.6. BP Model Prediction

To determine whether the $\alpha$ predictions of the RF and GSCV-RF models were superior, the passive support and active support BP models for $\alpha$ prediction were constructed using an $\alpha$ prediction indicator system. Using Equation (12), the number of nodes in the hidden layer is determined [26]:

$$n_h = 2I + 1 \tag{12}$$

where $n_h$ is the number of hidden layer nodes, and $I$ is the number of hidden layer nodes.

Table 7 displays the input parameters of the two BP prediction models. After the training, the passive and active support training sets were utilized to predict the $\alpha$ of the corresponding test sets, and the results are depicted in Figures 12 and 13.

As seen in Figures 12a and 13a, a number of samples were clustered near y = x, indicating that the actual measurement value of the passive and active support training sets and test sets had less correlation with the prediction value. In the passive support training sets, the values of *MAE*, *RMSE* and $R^2$ were 0.0111, 0.0136 and 0.8945, respectively, and in the passive support test sets, the values were 0.0127, 0.0182 and 0.8455, respectively. In the active support training sets, the values were 0.0038, 0.0050 and 0.7533, respectively, and in the active support test sets, the values were 0.0041, 0.0056 and 0.7235, respectively. All the values of the *MAE* and *RMSE* of datasets was less than 0.05. Besides, except for the passive support training sets, the value of $R^2$ for the remaining three datasets was less than 0.85.

As observed in Figure 12b,c and Figure 13b,c, the sample numbers of the passive support training sets were 2, 3, 4, . . . , 100 (42 samples in total). The deviation of the prediction from the field-measured value was slightly large, accounting for 42% of all the sample numbers. The sample numbers of the passive support test sets were 5, 6, 7, 8, 11, 12, 17 and 24. The deviation of the prediction from the field-measured value was slightly large, accounting for 33% of all the sample numbers. The sample numbers of the active support

training sets were 2, 5, 6, . . . , 30 (22 samples in total). The deviation of the prediction from the field-measured value was slightly large, accounting for 73.3% of all the sample numbers. The sample numbers of the active support test sets were 3, 5 and 6. The deviation of the prediction from the field-measured value was slightly large, accounting for 50% of all the sample numbers. In addition, the prediction value of the remaining samples was close to the actual value.

In Figures 12d and 13d, the frequency distribution of the samples' relative error is depicted. There were fewer samples near the 0 point and more samples further from it.

As stated previously, the BP model achieved poorer prediction results for the passive and active support datasets, and its prediction error for the samples was larger, the ratio of the larger error samples was higher, with the exception of the passive support training sets, the value of $R^2$ for the remaining sets was too low, indicating that the model's prediction accuracy was inferior.

**Table 7.** Input parameters of the BP model.

| No. | Parameter Name | Value (Passive Support) | Value (Active Support) |
|---|---|---|---|
| 1 | Number of nodes in the input layer | 7 | 4 |
| 2 | Number of nodes in the output layer | 15 | 9 |
| 3 | Number of nodes in the implicit layer | 1 | 1 |

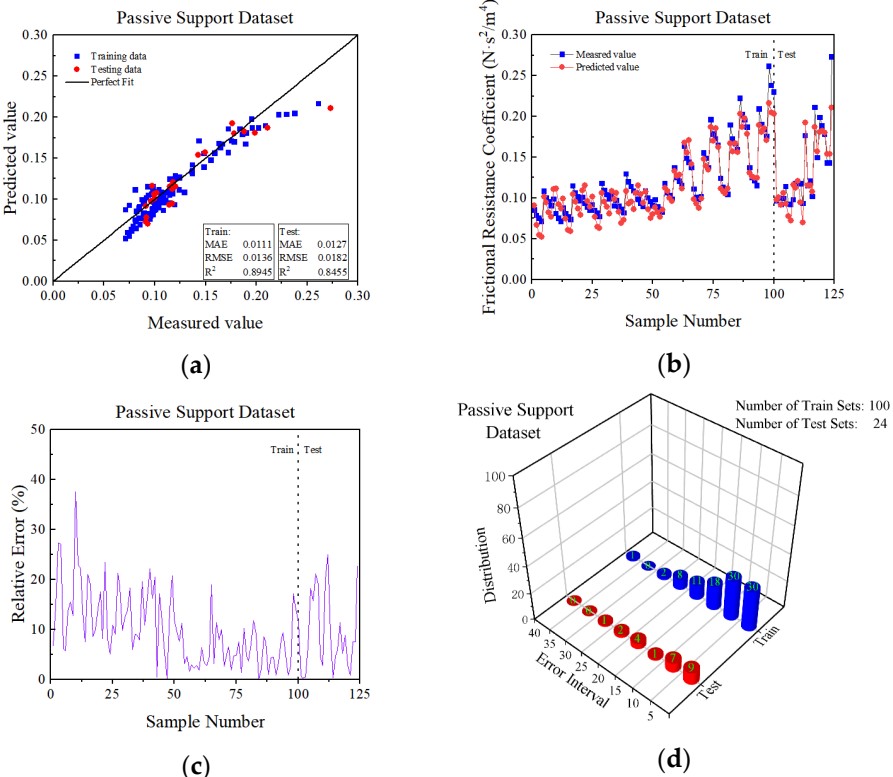

**Figure 12.** The passive support BP model prediction results. (**a**) Correlation evaluation of the measured and predicted values of $\alpha$; (**b**) curves of the measured and predicted values of the samples; (**c**) sample prediction error curve; (**d**) sample error frequency distribution.

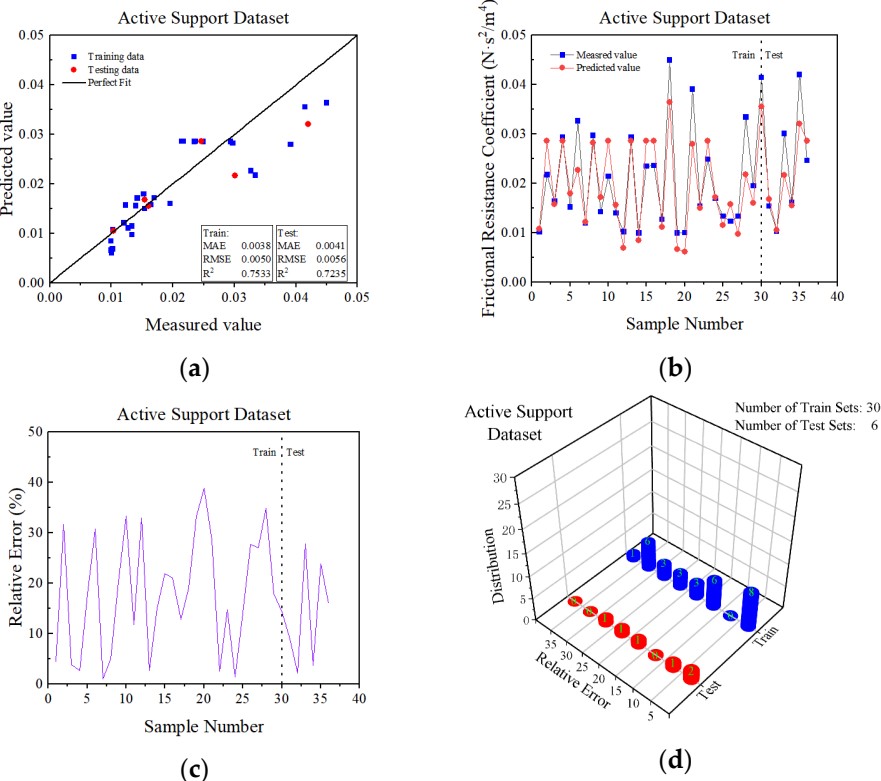

**Figure 13.** The active support BP model prediction results. (**a**) Correlation evaluation of the measured and predicted values of $\alpha$; (**b**) curves of the measured and predicted values of the samples; (**c**) sample prediction error curve; (**d**) sample error frequency distribution.

### 3.7. Prediction Result Comparison

By comparing Figures 8 and 11, we determined that the GSCV-RF model and the RF model had the same prediction tendency for the datasets. Their main difference was the accuracy of their prediction results. Table 8 displays the quantitative evaluation results of the RF and GSCV-RF models. With respect to the prediction results of the passive support training sets provided by the GSCV-RF model, the value of *MAE* decreased by 5.26%, the value of *RMSE* decreased by 13.79% and the value of $R^2$ increased by 0.13%. With respect to the prediction results of the passive support test sets provided by the GSCV-RF model, the value of *MAE* remained unchanged, the value of *RMSE* decreased by 1.26% and the value of $R^2$ increased by 0.35%. With respect to the prediction results of the active support training sets provided by the GSCV-RF model, the value of *MAE* increased by 40%, the value of *RMSE* increased by 26.67% and the value of $R^2$ decreased by 1.37%. Even though the rate of the increase in *MAE* and *RMSE* was greater, their respective indicator magnitude order was lower, and the error was still at a lower level. With respect to the prediction results of the active support test sets provided by the GSCV-RF model, the value of *MAE* decreased by 11.11%, the value of *RMSE* decreased by 9.68% and the value of $R^2$ increased by 1.41%. Consequently, we realized that, compared to the RF model, the GSCV-RF model is superior in its prediction ability, yielding more accurate and reliable data.

Comparing the results in Figures 10 and 13 and combining them with the quantitative evaluation results of the models in Table 8, we determined that the BP model has a larger prediction error and lower accuracy for the datasets. With respect to the prediction results of the passive support training sets provided by the BP model, the value of *MAE* increased by 516.67%, the value of *RMSE* increased by 444% and the value of $R^2$ decreased by 10.24%. With respect to the prediction results of the passive support test sets provided by the BP model, the value of *MAE* increased by 13.39%, the value of *RMSE* increased by 15.92% and the value of $R^2$ decreased by 4.41%. With respect to the prediction results of the active

support training sets provided by the BP model, the value of *MAE* increased by 171.43%, the value of *RMSE* increased by 163.16% and the value of $R^2$ decreased by 21.86%. With respect to the prediction results of the active support test sets provided by the BP model, the value of *MAE* increased by 70.83%, the value of *RMSE* increased by 100% and the value of $R^2$ decreased by 22.15%.

The RF model and GSCV-RF model offer the best prediction effects among the three models presented in this research, whereas the GSCV-RF model provides the most accurate $\alpha$ prediction. While the BP model has more substantial error samples and a lower $\alpha$ prediction accuracy. For the purpose of $\alpha$ prediction, the GSCV-RF model is the best of the three models, followed by the RF model and the BP model.

**Table 8.** Quantitative evaluation of the model results.

| Predictive Models | | *MAE* | *RMSE* | $R^2$ |
|---|---|---|---|---|
| RF | Passive Support Training Set | 0.0019 | 0.0029 | 0.9952 |
| GSCV-RF | | 0.0018 | 0.0025 | 0.9965 |
| BP | | 0.0111 | 0.0136 | 0.8945 |
| RF | Passive Support Test Set | 0.0112 | 0.0159 | 0.8814 |
| GSCV-RF | | 0.0112 | 0.0157 | 0.8845 |
| BP | | 0.0127 | 0.0182 | 0.8455 |
| RF | Active Support Training Set | 0.0010 | 0.0015 | 0.9775 |
| GSCV-RF | | 0.0014 | 0.0019 | 0.9641 |
| BP | | 0.0038 | 0.0050 | 0.7533 |
| RF | Active Support Test Set | 0.0027 | 0.0031 | 0.9165 |
| GSCV-RF | | 0.0024 | 0.0028 | 0.9294 |
| BP | | 0.0041 | 0.0056 | 0.7235 |

## 4. Conclusions

In this study, in order to solve problems such as the minute and complicated work and larger measurement errors in the $\alpha$ prediction of mines, we utilized an RF algorithm to build a prediction model that yields accurate and reliable $\alpha$ results. Because RF prediction results are more influenced by the super parameter, the GSCV algorithm was developed to optimize RF's hyperparameters, and the GSCV-RF model was constructed to predict $\alpha$. In order to determine whether or not RF was superior in $\alpha$ prediction, the authors constructed a BP model using a BP neural network, which is a common method of obtaining $\alpha$, and used it to make $\alpha$ predictions for the same datasets. The quantitative evaluation of each model's prediction results was illustrated by the values of *MAE*, *RMSE* and $R^2$, and a graphical representation of the relationship between the actual sample measurement value and the prediction value and errors was provided. Therefore, the conclusion of the paper is as follows:

1.  The paper began with the roadway support type, and after classifying the roadways as passive support or active support, the passive support $\alpha$ prediction indicator system and the active support $\alpha$ prediction indicator system were developed, respectively. The study demonstrated that the accuracy of these two support systems combined with machine learning, which can successfully predict $\alpha$, is dependent on the algorithm employed.

2.  The paper introduced the RF algorithm to solve the problem of $\alpha$ determination. To avoid the super parameter's influence, the GSCV algorithm was also introduced, and the GSCV-RF prediction model was constructed to predict the passive support training sets. The results were $MAE = 0.0018$, $RMSE = 0.0025$ and $R^2 = 0.9965$. In the prediction of the passive support test sets, the results were $MAE = 0.0112$, $RMSE = 0.0157$ and $R^2 = 0.8845$. In the prediction of the active support training sets, the results were $MAE = 0.0014$, $RMSE = 0.0019$ and $R^2 = 0.964$. In the prediction of the active support test sets, the results were $MAE = 0.0024$, $RMSE = 0.0028$ and $R^2 = 0.9294$. The smaller

*MAE* and *RMSE*, as well as the larger $R^2$, demonstrated that the GSCV-RF model can produce more accurate and reliable predictions of $\alpha$.

3. After comparing and analyzing the three models, we concluded that the GSCV-RF model was superior in $\alpha$ prediction, followed by the RF model and the BP model. The BP model's $R^2$ was too low, proving that the GSCV-RF model was superior in $\alpha$ prediction.

Therefore, the GSCV-RF model is able to avoid the tedious and time-consuming work involved in filed measurements and to obtain accurate and reliable $\alpha$ results. In the design of ventilation systems, the GSCV-RF model should be utilized to keep ventilation systems in a safe and reasonable state.

**Author Contributions:** Conceptualization, C.G. and X.W.; methodology, C.G. and X.W.; software, C.G. and D.H.; validation, C.G.; data curation, H.L. and M.J.; writing—original draft preparation, C.G.; writing—review and editing, C.G., X.W. and J.L.; visualization, C.G. and Y.Z.; funding acquisition, X.W. and J.L. All authors have read and agreed to the published version of the manuscript.

**Funding:** This research was funded by the Kunming University of Science and Technology Introduced Talent Research Startup Fund Project (No. KKSY201721032) and the Key Research and Development Project of Yunnan Province (No. 202003AC100002).

**Institutional Review Board Statement:** Not applicable.

**Informed Consent Statement:** Not applicable.

**Data Availability Statement:** The data presented in this study are available in the References [8,24,25].

**Conflicts of Interest:** The authors declare no conflict of interest.

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
