# Peer review of "Predicting the Mine Friction Coefficient Using the GSCV-RF Hybrid Approach"

_applsci, doi:10.3390/app122312487_

Round 1
Reviewer 1 Report
The manuscript provides comparison between RF and GSCV-RV for predicting the mine friction coefficient. The manuscript is recommended for publication considering the following comments:
1. Proofreading is highly recommended to enhance the readability. Avoid very long sentences, for instance: page 2 lines 72 – 78. This one sentence could be splitted.
2. In page 2 line 71: ‘’ … their proposed model have defects.’’. It is recommended to delete this phrase and rewrite the preceding sentences (lines 68-70). It looks like tough and rush critics rather than describing the literature.
3. In page 7, Section 3.1: How the 124 (resp. 22) sets were selected for the passive and active sets. Please discuss.
4. There is contradiction of the lower limits in the hyperparameters between Eq.(9) and Table 3. Also, the step length for max_features is 0.1, while it was assumed to be 1.0, please see line 146.
5. Tables 4, column 3 should be optimized value but not range. How the step length was optimized.
6. Random selection of training and testing sets could yield different results. To fairly compare the efficiency of the models, the data set division of the different models SHOULD be the same.
7. In the line of this study, it is recommended to cite and refer to recent paper on optimize the architecture of ML and its hyperparameters. Please refer to; Neural Computing and Applications (2021), 33 (6), 1923-1933 DOI: https://doi.org/10.1007/s00521-020-05035-x
Author Response
Thank you for your valuable comments. Please see the attachment.

Reviewer 2 Report
In the present study, the authors have claimed that, this paper discusses about the GSCV-RF model prediction of the friction resistance coefficient, and the model was constructed by the passive and the active support mode by the GSCV algorithm optimizing the RF algorithm. All in all the research topic is interesting but the authors did not organize well, especially in the “Materials and Methods” and the Results and Discussions” sections. For all reasons given above and the following comments, I invite the authors to in-depth revise their manuscript and submit again for re-review. I expect they accurately answer the comments and addressed them in the main text.
Comments:
1. The title of the manuscript may be revised as: “Predicting the mine friction coefficient using GSCV-RF hybrid approach”.
2. In abstract, the authors need to have distinct three parts: i) what topic they are going to investigate, ii) which method and assumptions are implemented, and iii) finally, what is the important results.
3. It is recommended to summarize and modify the literature review. Following references is also recommended to expand the predictive models description.
- “Predicting Angle of Internal Friction and Cohesion of Rocks Based on Machine Learning Algorithms”. Mathematics. (2022): 10, 3875. https://doi.org/10.3390/math10203875
Predicting Angle of Internal Friction and Cohesion of Rocks Based on Machine Learning Algorithms. Mathematics 2022, 10, 3875.
- "Estimation of Splitting Tensile Strength of Modified Recycled Aggregate Concrete Using Hybrid Algorithms." Steel and Composite Structures (2022): Vol. 44, No. 3, pp 375-392. https://doi.org/10.12989/scs.2022.44.3.375
- "Predicting resilient modulus of flexible pavement foundation using extreme gradient boosting based optimised models." International Journal of Pavement Engineering (2022): 1-20. https://doi.org/10.1080/10298436.2022.2095385
4. The most critical concerns raised from the manuscript is about the utilized dataset. For the passive support and active support predictions, 124 and 22 dataset have been utilized, respectively. My best knowledge from the literature review in that, no prediction models cannot developed using 24 data set. If so, I strongly recommend to extend the dataset in order to develop more reasonable predictive models.
5. Other wrong strategy is that, the authors divided the training and testing data without any study and logical justifications. They selected 80% and 75% for 124 and 22 dataset, respectively, which I think it cannot lead to train an accurate model. Usually, for the smaller number of data, more contribution of training data should be adopted, provided that it does not lead to the "overtraining" of the model.
6. For both of the passive support and active support dataset, statistical specifications table (including: min, max, mean, median, count, stdev, skewness, kurtosis and etc.) should be provided separately for the training and testing data.
7. There is no reasonable discussions about the statistical analysis of the dataset. This may be done using Pearson correlation plot, boxplot, violin plot, frequency distribution plot and etc.
8. The optimized amount of internal hyper-parameters of each model should be presented in a table.
9. The fitness curve for the iterations should be demonstrated for each models (single and hybrid) considering R2 or RMSE.
10.The plot of predicting data against the measured data should be presented for both of the training and testing data. Training and testing data can be distinguished by the legend.
11.The plot of the error distribution (or residual distribution) and the frequency distribution of measured and predicted values should be presented for the training and testing datasets.
12.It is needed to re-write the "conclusion" considering: i) Restate the thesis, ii) Reiterate supporting points, iii) Make a connection between opening and closing statements, and finally, iv) Provide some insight
Author Response

(The authors gave the same response as above.)

Reviewer 3 Report
The work provides a prediction algorithm to identify the friction coefficient in mines. The presented algorithm is a combination of the Grid Search Cross Validation and the Rain Forest methods, obtaining a satisfactory performance for the set of data analyzed in the manuscript. The following comments are given to the authors:
1. The work gives an interesting result which can be useful on the mining industry. However, a brief introduction of the physical meaning of the friction coefficient and its importance in the calculation of mine ventilation systems can be useful for readers interested in apply your algorithms in a practical application.
2. A real mine is an evolving system, there are different that can affect the value of the friction coefficient. Irregularities in the profile of the ducts, changes of cross sectional area, pollution of the air with extracted minerals, CO2 in the environment, etc., are factors that can affect the value of the friction coefficient. How can your algorithm handle such disturbances?
3. The proposed algorithm is compared in Table 6 with the Rain Forest method. Since the proposition is a combination of RF and GSCV methods, it seems to be clear to obtain better results. A comparison with different prediction models could be useful to highlight the performance of your proposition.
Author Response

(The authors gave the same response as above.)

Round 2
Reviewer 1 Report
The manuscript has been revised. It is recommended for the publication
Reviewer 2 Report
Valuable efforts have been conducted and the authors have responded and addressed all of my comments carefully and I am satisfied with the modifications. The manuscript is ready to publish in the present form.